# Purifying Selection Influences the Comparison of Heterozygosities between Populations

**DOI:** 10.3390/biology13100810

**Published:** 2024-10-10

**Authors:** Sankar Subramanian

**Affiliations:** Centre for Bioinnovation, School of Science, Technology, and Engineering, The University of the Sunshine Coast, Moreton Bay, QLD 4502, Australia; ssankara@usc.edu.au; Tel.: +61-7-5430-2873; Fax: +61-7-5430-2881

**Keywords:** heterozygosity, nucleotide diversity, purifying selection, genetic drift, effective population size, small populations

## Abstract

**Simple Summary:**

Heterozygosity is a fundamental measure used to compare the level of genetic variation between populations. However, such a comparison is influenced by the magnitude of selection pressure on the genomic regions used. Using the genomic regions free from selection, the heterozygosity of large populations was two times higher than that of small populations. However, this difference was only ~1.6 times for the heterozygosities estimated using the regions under selective constraints. This suggests an excess in the heterozygosities of constrained regions due to the presence of deleterious variants in small populations. This investigation found a correlation between the effective population size and the magnitude of the excess in the constrained-region diversity. Furthermore, the excess was found to be much higher for highly constrained genes and lower for genes under relaxed selective constraints. The excess was also higher for highly expressed genes compared to those with low expression levels. These results emphasize the use of neutral regions, less constrained genes, or lowly expressed genes when comparing the heterozygosities between populations.

**Abstract:**

Heterozygosity is a fundamental measure routinely used to compare between populations to infer the level of genetic variation and their relative effective population sizes. However, such comparison is highly influenced by the magnitude of selection pressure on the genomic regions used. Using over 2 million Single Nucleotide Variants (SNVs) from chimpanzee and mouse populations, this study shows that the heterozygosities estimated using neutrally evolving sites of large populations were two times higher than those of small populations. However, this difference was only ~1.6 times for the heterozygosities estimated using nonsynonymous sites. This suggests an excess in the nonsynonymous heterozygosities due to the segregation of deleterious variants in small populations. This excess in the nonsynonymous heterozygosities of the small populations was estimated to be 23–31%. Further analysis revealed that the magnitude of the excess is modulated by effective population size (*N_e_*) and selection intensity (*s*). Using chimpanzee populations, this investigation found that the excess in nonsynonymous diversity in the small population was little (6%) when the difference between the *N_e_* values of large and small populations was small (2.4 times). Conversely, this was high (23%) when the difference in *N_e_* was large (5.9 times). Analysis using mouse populations showed that the excess in the nonsynonymous diversity of highly constrained genes of the small population was much higher (38%) than that observed for the genes under relaxed selective constraints (21%). Similar results were observed when the expression levels of genes were used as a proxy for selection intensity. These results emphasize the use of neutral regions, less constrained genes, or lowly expressed genes when comparing the heterozygosities between populations.

## 1. Introduction

Heterozygosity is the measure of genetic variation in a population, which is determined by the product of effective population size (*N_e_*) mutation rate (*μ*) [1]. However, the heterozygosity of different populations of the same species is largely determined by *N_e_*, as the mutation rate does not vary significantly among populations. Therefore, large populations have higher heterozygosity than small ones. On the other hand, purifying selection removes mildly deleterious mutations from constrained genomic regions. This results in reduced heterozygosity in these regions compared to those under neutral evolution [2]. Furthermore, the strength of selection is weak in small populations owing to strong genetic drift. Therefore, a much higher fraction of more mildly deleterious variants is expected to segregate in small populations compared to large ones [3]. A number of previous studies provide empirical proof for this prediction [4,5,6,7,8,9,10].

The ratio of heterozygosities at nonsynonymous-to-synonymous sites (*π_N_*/*π_S_*) is routinely used to compare the deleterious mutation loads between populations. A high ratio suggests an elevated fraction of segregating nonsynonymous variants in a population. Previous studies showed that the reduction in the population size results in a high *π_N_*/*π_S_* ratio. The higher *π_N_*/*π_S_* ratio observed in Neanderthals than that in modern humans was due to the drastic decline in the population of Neanderthals before their extinction [4,5]. Similarly, the long-term reduction in population size appears to have led to a high *π_N_*/*π_S_* ratio in some of the populations of great apes [6]. Earlier studies estimated the *π_N_*/*π_S_* ratio in bottlenecked island populations and compared them with their mainland counterparts. The island fox populations were found to have a much higher *π_N_*/*π_S_* ratio than that of grey fox—their mainland counterpart [7]. A similar result was reported for the comparison of Wrangel Island and mainland (Oimyakon) mammoths [8]. An earlier study based on the comparison of island and mainland populations belonging to over 11 species of Passerine birds showed an elevated *π_N_*/*π_S_* ratio in all island populations [11]. Furthermore, the landlocked salmon population was found to have a much higher *π_N_*/*π_S_* ratio than the anadromous salmon populations of Norway [9].

During the process of domestication of wild animals, only a small number of founders were used, which also resulted in a significant reduction in the effective population size [12]. Hence, domesticated populations tend to have more deleterious variants than their wild relatives. Therefore, previous studies compared the *π_N_*/*π_S_* ratios of domesticated animals and their wild progenitors. These studies observed a much higher *π_N_*/*π_S_* ratio in domesticated populations such as dogs than wolves [13]. Similar higher ratios were reported for other domesticated animals such as cow, yak, pig, and silkworm [14,15] and domesticated plants such as rice and sunflower [16,17] in comparison with those of their wild counterparts. Furthermore, the intensity of breeding also appears to contribute to the accumulation of deleterious variants. The *π_N_*/*π_S_* ratios of the intensely bred taurine cows were higher than those estimated for the indicine breeds [18].

The high *π_N_*/*π_S_* ratios observed for the bottlenecked, island, and domesticated populations clearly suggest a higher fraction of nonsynonymous variants in these populations compared to the unrestricted, mainland, and wild counterparts, respectively. Therefore, these excess nonsynonymous variants spuriously contribute to higher nonsynonymous diversity in small populations. The comparison of *π_N_*/*π_S_* ratios provided only the qualitative difference (high and low) between the large and small populations compared. However, it is important to quantify the excess in the nonsynonymous diversity or constrained-site diversity of small populations. For this purpose, the whole-genome data from two populations of mice and four populations of chimpanzees were used. These populations have widely different *N_e_* values, which were estimated by previous studies [6,19]. First, the diversities in neutral and constrained genomic regions of large and small populations were compared, and then, using these two quantities, the excess in the small populations was calculated. This study also examined the influence of effective population size (*N_e_*) and selection intensity (*s*) on the rate of overestimation. For *N_e_*, the excess fraction of nonsynonymous diversities was obtained from the comparisons involving three large and small pairs of chimpanzee populations. For *s*, the excess in the small mouse population for the genes under different levels of selective pressure was compared. Previous studies showed that the level of gene expression correlates with the magnitude of selection pressure [20,21]. Therefore, gene expression was used as a proxy for the measure of selection constraint; the excess in the nonsynonymous diversity of the small mouse population was calculated, which was then compared between highly and lowly expressed genes.

## 2. Methods

### 2.1. Quantifying the Excess Fraction of Deleterious Variants Segregating in Small Populations

The classical McDonald–Kreitman test was based on the assumption that the ratio of nonsynonymous-to-synonymous divergence (*d_N_*/*d_S_*) is equal to the ratio of nonsynonymous-to-synonymous diversity (or polymorphisms) (*π_N_*/*π_S_*) [22]. While a higher *d_N_*/*d_S_* ratio suggests adaptive evolution, a higher *π_N_*/*π_S_* ratio suggests purifying selection due to the segregation of deleterious alleles. The latter was investigated by many studies using the mitochondrial and nuclear genes [23,24,25,26]. Using the similar assumption, we can predict that under neutrality, the ratio of diversities at constrained-to-neutral sites of a large population (*π_CL_*/*π_NL_*) is equal to the ratio of these diversities of a small population (*π_CS_*/*π_NS_*) of the same species.
$$\frac{\pi_{CL}}{\pi_{NL}}=\frac{\pi_{CS}}{\pi_{NS}}$$

However, a number of previous studies have shown that the ratio of constrained-to-neutral site diversities of small populations was always higher than that of large populations [4,5,6,7,8,9,10,13,14,15,16,17,18]. This can be expressed as
(1)$$\frac{\pi_{CL}}{\pi_{NL}}<\frac{\pi_{CS}}{\pi_{NS}}$$
where $\frac{\pi_{CL}}{\pi_{NL}}$ and $\frac{\pi_{CS}}{\pi_{NS}}$ are the ratios of diversities at constrained and neutral sites of large and small populations, respectively. This relationship assumes that the fraction of sites under positive selection is very small (e.g., in vertebrates).

By subtracting the excess fraction of diversity at constrained sites of the small population, these two ratios will become equal. This can be written as
$$\frac{\pi_{CL}}{\pi_{NL}}=\frac{{\pi_{CS}-\delta \pi }_{CS}}{\pi_{NS}}$$
where *δ* is the fraction of excess polymorphisms contributing to heterozygosity in small populations.

By rearranging the above equation, we obtain
$$\frac{\pi_{CL}}{\pi_{NL}}=\frac{\pi_{CS}}{\pi_{NS}}-\delta \frac{\pi_{CS}}{\pi_{NS}}$$

This can be simplified as
$$\omega_{L}=\omega_{S}-\delta \omega_{S}$$
where $\omega_{L}=\frac{\pi_{CL}}{\pi_{NL}}$ and $\omega_{S}=\frac{\pi_{CS}}{\pi_{NS}}$.
$$\omega_{L}=\omega_{S}(1-\delta )$$

After simplifications, we obtain
(2)$$\delta =1-\frac{\omega_{L}}{\omega_{S}}$$

The excess fraction of constrained-site heterozygosity in small populations can be expressed in percentage as
(3)$$\delta \left(\%\right)=\frac{{\omega_{S}-\omega }_{L}}{\omega_{S}}\times 100$$

The above equation could be used to calculate overestimation for any type of genomic regions. For instance, diversity in exomes or nonsynonymous sites could be substituted for *π_C_* and the diversity of the whole genome, intron, synonymous or intergenic sites could be substituted for *π_N_*.

### 2.2. Genome and Gene Expression Data

The whole-genome data for *M. m. castaneous* (11) and *M. m. musculus* (11) were generated by a previous study [19] and obtained from the Dryad digital repository (https://doi.org/10.5061/dryad.66t1g1k1j, accessed on 29 September 2023). Although this study sequenced many genomes from these populations, only the individuals who did not have admixture from any other populations were selected. The complete genome data of four chimpanzee populations—*Pan troglodytes ellioti* (10), *Pan troglodytes schweinfurthii* (6), *Pan troglodytes troglodytes* (4), and *Pan troglodytes verus* (5)—were obtained from a previous study [6] and the whole-genome data are available at https://eichlerlab.gs.washington.edu/greatape/data/VCFs/SNPs/ (accessed on 29 September 2023). This study included only the autosomal Single Nucleotide Variants (SNVs) with two alleles in the analyses. Using the software *SnpEff* (v4_1d), the genomes were annotated to find synonymous, nonsynonymous, intergenic, and intron SNVs [27]. Furthermore, using the genomic coordinates of exon boundaries, the protein-coding genes were extracted. The program PAML was used to extract the number of synonymous and nonsynonymous sites [28]. The final dataset used in this study contained 2.8 and 4.8 million SNVs for chimpanzee and mouse populations, respectively.

### 2.3. Data Analysis

To determine the intensity of selection on each SNV, the *PhyloP* score was used [29]. The base-wise *PhyloP* scores based on the multiple sequence alignment of 59 vertebrates were obtained from the UCSC genome browser (https://hgdownload.soe.ucsc.edu/goldenPath/mm10/phyloP60way/ obtained on 7 March 2019). The proportion of sites in a gene having a *PhyloP* score > 2.0 was then calculated. The proportion of constrained sites provided a measure of selection pressure on each gene. The final dataset contained 14,870 genes. The 14,870 were grouped into 15 categories based on the proportion of conserved sites (1000 genes per category). The large-scale microarray array data from 61 mouse tissues were obtained, for which the log-transformed mean level of expression was available [30]. The gene expression levels for the protein-coding genes were available for 9089 genes, which were grouped into nine categories based on the mean expression level (1000 genes per category). In-house Perl scripts were used to perform all data analyses.

### 2.4. Statistical Analysis

The heterozygosity was estimated using Tajima’s mean pairwise differences per site method using the whole genome and intron, intergenic, synonymous, and nonsynonymous sites [31]. The standard deviation was estimated using the bootstrap method (1000 replicates) by resampling 1 Mb genomic blocks. The nature of the relationship (e.g., linear or nonlinear) between the proportion of constrained sites or gene expression level and the magnitude of excess is not known, and the type of distribution (normal or other) of these variables is also unknown. Therefore, non-parametric Spearman rank correlation was used to test the strength of the relationship.

## 3. Results

### 3.1. The Excess in the Constrained-Site Diversity of Small Populations

To compare the heterozygosities of large and small populations, the whole-genome data of two populations of mice and four populations of chimpanzees were used, which contained 4.8 and 2.8 million SNVs (or segregating sites), respectively. The populations of mouse (*Mus musculus castaneous* and *Mus musculus musculus*) and chimpanzee (*Pan troglodytes troglodytes*, *Pan troglodytes schweinfurthii*, *Pan troglodytes ellioti*, and *Pan troglodytes verus*) have widely different *N_e_* values. Previous studies estimated the population sizes of *M. m. castaneous* and *M. m. musculus* to be 200,000 to 400,000 and 60,000–120,000, respectively [32,33,34,35]. The *N_e_* values of *P. t. troglodytes*, *P. t. schweinfurthii*, *P. t. ellioti* and *P. t. verus* were 60,000, 24,000, 18,000, and 10,000, respectively [6]. The heterozygosity was estimated for the whole genome and, exome, intron, intergenic, nonsynonymous, and synonymous sites for the populations of mice and chimpanzees. The ratios of these estimates for *M. m. castaneous* (large) and *M. m. musculus* (small) and *P. t. troglodytes* (large) and *P. t. verus* (small) populations are shown in Figure 1A. The ratio of diversities obtained for the neutral regions, including those from the genome, intron, intergenic, and synonymous sites, are higher than those observed for constrained sites such as exomes and nonsynonymous sites. For example, the intron diversity of *M. m. castaneous* was 2.26 times higher than the intron diversity estimated for *M. m. musculus*. In contrast, the nonsynonymous diversity of the former was only 1.56 times higher than the nonsynonymous diversity observed for the latter. Similarly, the intron diversity of *P. t. troglodytes* was 2.1 times higher than the value estimated for *P. t. verus*, but this difference was only 1.62 times for the nonsynonymous diversity. Using Equation (3), the excess was calculated by substituting the exome diversity of *M. m. castaneous* and *M. m. musculus* for *π_CL_* and *π_CS_*, respectively, and substituting intron diversity for *π_NL_* and *π_NS_*, respectively. The excess in the exome diversity (compared to that observed for neutral sites) of *M. m. musculus* was computed to be 13.6% (Figure 1B). A similar analysis using the exome and intron diversities of *P. t. troglodytes* and *P. t. verus* revealed a 16.7% excess in the exome diversity of *P. t. verus*. Then, the excess was calculated by substituting the nonsynonymous diversities for *π_CL_* and *π_CS_* and intron diversities for *π_NL_* and *π_NS_* for large and small populations, respectively. This revealed an excess of 31.0% and 23.2% in the nonsynonymous diversities of *M. m. musculus* and *P. t. verus,* respectively.

### 3.2. The Magnitude of the Excess Fraction Correlates with Effective Population Size (N_e_)

To examine the influence of *N_e_* on the excess fraction of constrained-site diversity, the nonsynonymous and intron diversities of four populations of chimpanzees were compared. The *N_e_* of *P. t. troglodytes* was the largest (60,000), and hence it was used as a reference. The ratios of intron diversities of *troglodytes*/*schweinfurthii*, *troglodytes*/*ellioti* and *troglodytes*/*verus* (Figure 2A) were then calculated. These ratios correlated with the difference in *N_e_*. The ratios of intron diversities of the above-mentioned pairs were 1.07, 1.17, and 1.98 times, and the corresponding ratios of *N_e_* of each pair were 2.4, 3.4, and 5.9 times, respectively. However, the correspondence between the ratios of nonsynonymous diversities and the ratio of *N_e_* was not very clear. The excess fraction of nonsynonymous diversity for each pair was computed, which showed an increasing trend with the increasing ratio of *N_e_* (Figure 2B). For example, *Troglodytes*/*Schweinfurthii* have the least difference (2.4 times) between their *N_e_* values (60,000 and 25,000, respectively) and the excess in the nonsynonymous diversity of *P. t. schweinfurthii* was only 5.7%, whereas the difference in the *N_e_* of the pair *troglodytes*/Elliott (60,000 and 18,000, respectively) was intermediate (3.4 times), and the excess fraction of nonsynonymous diversity of *P. t. ellioti* was also intermediate (12.7%). The difference in the *N_e_* of the *troglodytes*/*verus* pair (60,000 and 10,000, respectively) was the highest (5.9 times), and the excess fraction of nonsynonymous diversity of *P. t. verus* was also the maximum (23%) among these comparisons.

### 3.3. Positive Relationship between the Magnitude of the Excess and Selection Intensity (s)

The influence of selection pressure on the magnitude of the excess fraction of constrained-site diversity was investigated by using genes evolving under different magnitudes of selection. The proportion of sites in a gene with a *PhyloP* score > 2.0 was calculated and genes were grouped into 15 categories based this, and the nonsynonymous diversities of *M. m. castaneous* and *M. m. musculus* for each group of genes were estimated. Figure 3A shows a huge difference in the diversities of the two populations for genes with a much smaller proportion of constrained sites (<0.1 or <10%). The trend of this graph shows a reduction in the difference between the two diversities with the increase in the proportion of constrained sites. The nonsynonymous diversities of the two populations appear to be very similar for genes with >60% constrained sites. On the other hand, the difference in the intron diversities of the two populations was very similar across the groups of genes with different proportions of constrained sites (Figure 3B).

Using Equation (3), the excess proportion of nonsynonymous diversity of *M. m. musculus* was calculated for the genes belonging to each selective constraint category. Then, the proportion of constrained sites was plotted against the magnitude of excess in the nonsynonymous diversity of *M. m. musculus*. This showed a highly significant positive relationship (*r* = 0.83, *p* = 0.00019) between the two variables. The excess fraction was only 21% for genes under relaxed selection constraints (with <10% constrained sites). However, it was 38% for highly constrained genes (with >60% constrained sites), which is 83% higher than that observed for the relaxed genes. To measure the magnitude of the excess proportions for genes with different levels of expression, the protein-coding genes were grouped into nine categories based on their mean level of gene expression. An analysis revealed a highly significant positive correlation (*r* = 0.90, *p* = 0.002) between the level of gene expression and the magnitude of the excess (Figure 4B). The excess in nonsynonymous diversity of *M. m. musculus* was 42% for highly expressed genes (>7.0), and it was only 21% for the lowly expressed genes (<4.7).

## 4. Discussion

Previous studies showed that the ratio of nonsynonymous-to-synonymous diversity (*π_N_*/*π_S_*) estimated in within-species comparison was much higher than the nonsynonymous-to-synonymous divergence (*d_N_*/*d_S_*) estimated for between-species comparison [23,24,25]. The elevated *π_N_*/*π_S_* ratio was attributed to the excess of nonsynonymous polymorphisms segregating in populations that were mildly deleterious in nature. Later studies observed a higher *π_N_*/*π_S_* ratio in small or bottlenecked populations compared to large populations owing to the similar excess of the mildly deleterious nonsynonymous variants in small populations [4,5,6,7,8,9,10,13,14,15,16,17,18]. The excess fraction of segregating variants results in a spurious inflation of nonsynonymous heterozygosity of small populations. This is spurious because the elevated fraction of segregating deleterious variants in small populations is only transient, as they do not reach high frequencies and are removed by selection over time [3]. Furthermore, this excess fraction of harmful variants will also falsely inflate the effective population size inferred from this. Therefore, by comparing the neutral and constrained-site heterozygosities, the present study quantified the excess in the constrained-site heterozygosity in small populations with respect to the large populations. The results from small and large populations of chimpanzees and mice showed that the excess fraction was much higher for the nonsynonymous sites (23–31%) than that of exomes (14–17%). This is because the selection pressure is much higher on nonsynonymous sites compared to that on exomes. The observed similarity in the results from primates and rodents (Figure 1) suggests that the observed excess proportion of deleterious variants is universal, at least across mammals.

The results from chimpanzee populations showed that the excess fraction was the highest for the population with the smallest *N_e_* (e.g., 10,000 for *p. t. verus*). This is due to the strongest influence of genetic drift in this population, which leads to a much larger fraction of the excess deleterious variants segregating as the selection is too weak to remove them. In contrast, genetic drift was not so strong in the population with relatively large *N_e_* (e.g., 25,000 for *P. t. schweinfurthii*), and this leads to a small fraction of such excess deleterious variants segregating as selection would have been relatively efficient to remove most of them. The method described in this study will only capture the excess segregating fraction contributing to the constrained-site diversity of the small population in comparison with that of the large population. Therefore, the magnitude of the excess fraction depends on the difference between the *N_e_* of the large and small populations compared. A previous study suggested that the bias in sampling by selecting individuals from closely related or subdivided populations could also produce a low *N_e_* and high *π_N_*/*π_S_* ratio [36]. Although this is a possibility, it is unlikely such a bias could have occurred in all pairs of comparisons from both chimpanzee and mouse populations, as shown in this study.

The results of this study also revealed a positive correlation between the proportion of constrained sites and the magnitude of the excess fraction of nonsynonymous heterozygosity in small populations. This suggests that the fraction of excess is much higher for highly constrained genes than those under less selective constraints. This is because genes under high purifying selection have a large fraction of their sites under purifying selection, which is evident from the high proportion of sites with *PhyloP* scores > 2. Therefore, a large number of deleterious variants are expected to segregate in these genes. This leads to a high proportion of excess nonsynonymous diversity of small populations for these genes. In contrast, genes under relaxed selective constraints have less constrained sites and thus fewer segregating deleterious variants. Therefore, the excess fraction of nonsynonymous diversity of small populations was minimal for these genes. The level of gene expression was found to correlate negatively with the selection intensity of genes [20,21]. This means highly expressed genes are under high purifying selection. Therefore, the expression level of genes was used as the proxy for the intensity of selection pressure. The observed positive correlation between gene expression levels and the magnitude of excess in the nonsynonymous diversity of small populations independently confirms the inference from the previous analysis based on the proportion of constrained sites.

This study chose intron rather than synonymous sites as the proxy for neutral sites as the latter is known to be under weak selective constraints [37,38]. However, similar results were observed using synonymous sites to quantify the excess fraction (Appendix A). A well-documented phenomenon called background selection causes a reduction in the diversity of the neutral sites flanking highly constrained sites [39,40,41,42]. Previous studies showed that the effect of background selection is negligible on the neutral sites far away from protein-coding genes [41,42]. Therefore, the heterozygosity of intergenic sites that are 1 Mb away from protein-coding genes was estimated, which was then used to calculate the excess fraction in mouse and chimp populations. This analysis also produced results that were very similar to those observed using introns (Appendix A). The gene-based analysis could not be performed as the intergenic diversity (under the above-mentioned criteria) could not be calculated for many genes.

## 5. Conclusions

By comparing the *π_N_*/*π_S_* ratios of small and large populations, previous studies qualitatively inferred the excess in the nonsynonymous heterozygosity of the latter. The present study quantified this excess fraction of constrained-site heterozygosity of the small population. The *π_N_*/*π_S_* ratios vary between the populations of different species and hence cannot be compared across species. For example, the *π_N_*/*π_I_* ratios of *M. m. castaneous* and *M. m. musculus* are 0.15 and 0.22 but those for *P. t. troglodytes* and *P. t. verus* are 0.32 and 0.41. Although in both cases, the small populations have high ratios, their magnitude cannot be compared between species. However, using the method reported here, the excess fraction of nonsynonymous diversity of *M. m. musculus* is 31% and that of *P. t. verus* is 23%. This clearly suggests that the excess fraction of nonsynonymous polymorphisms is higher in *M. m. musculus* than in *P. t. verus*. Therefore, future studies could use this method to quantify the excess variants segregating in constrained sites of small populations and compare them across the populations from different species. The results of this study also suggest using neutral regions, alongside less constrained and lowly expressed genes, to compare the heterozygosity between populations.

## Figures and Tables

**Figure 1 biology-13-00810-f001:**
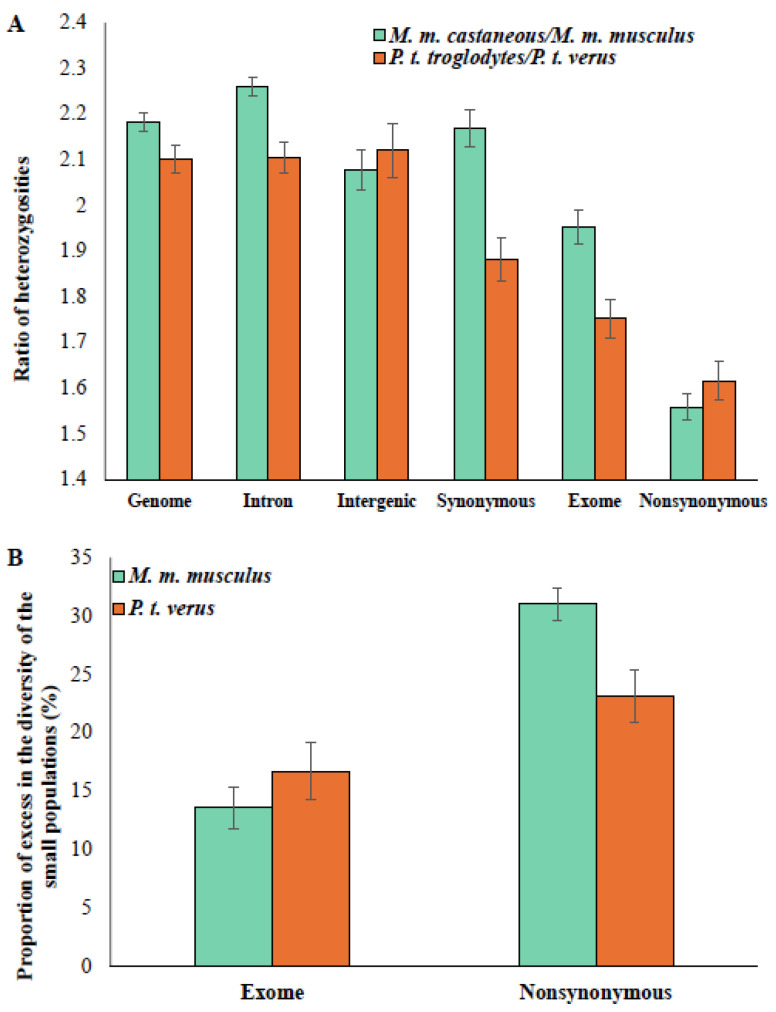
(**A**) Heterozygosity of various genomic regions was estimated for large and small populations of mice and chimpanzees. The ratios of heterozygosity obtained for *M. m. castaneous*/*M. m. musculus* and *P. t. troglodytes*/*P. t. verus* are shown. Error bars show the standard error of the mean. (**B**) The observed excess in the diversities of exome and nonsynonymous sites for *M. m. musculus* and *P. t. verus* is shown.

**Figure 2 biology-13-00810-f002:**
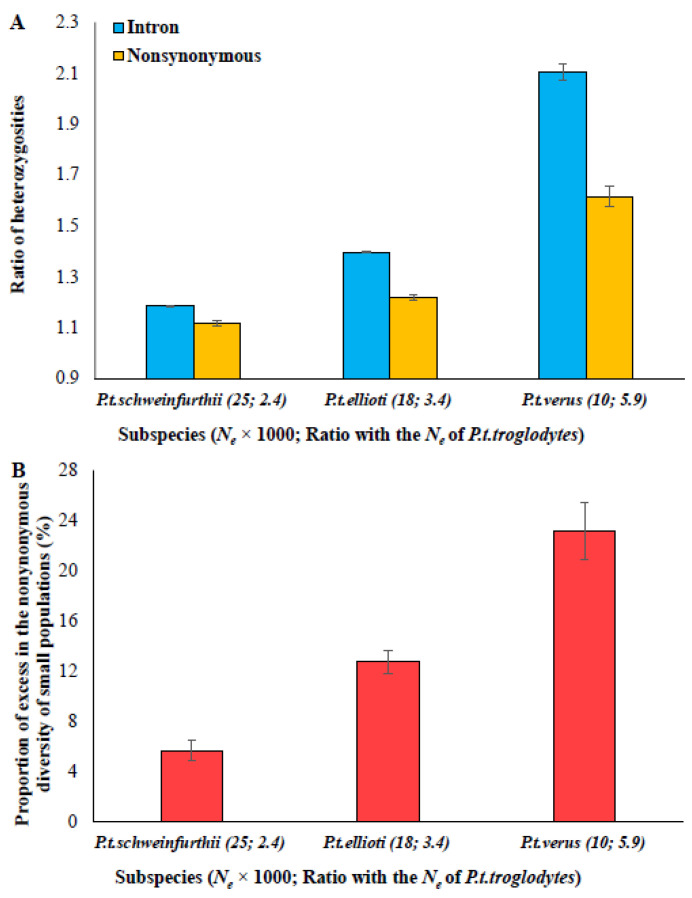
(**A**) The ratios of heterozygosities obtained using intron and nonsynonymous sites for the following pairs are shown: *P. t. troglodytes*/*P. t. schweinfurthii*, *P. t. troglodytes*/*P. t. ellioti* and *P. t. troglodytes*/*P. t. verus*. The *N_e_* of *P. t. troglodytes* is 60,000. The *N_e_* of small populations and the ratio of *N_e_* for the above-mentioned pairs are given in parentheses. (**B**) The fraction of excess in the nonsynonymous heterozygosities of *P. t. schweinfurthii*, *P. t. ellioti* and *P. t. verus* is shown. The error bars denote the standard error of the mean.

**Figure 3 biology-13-00810-f003:**
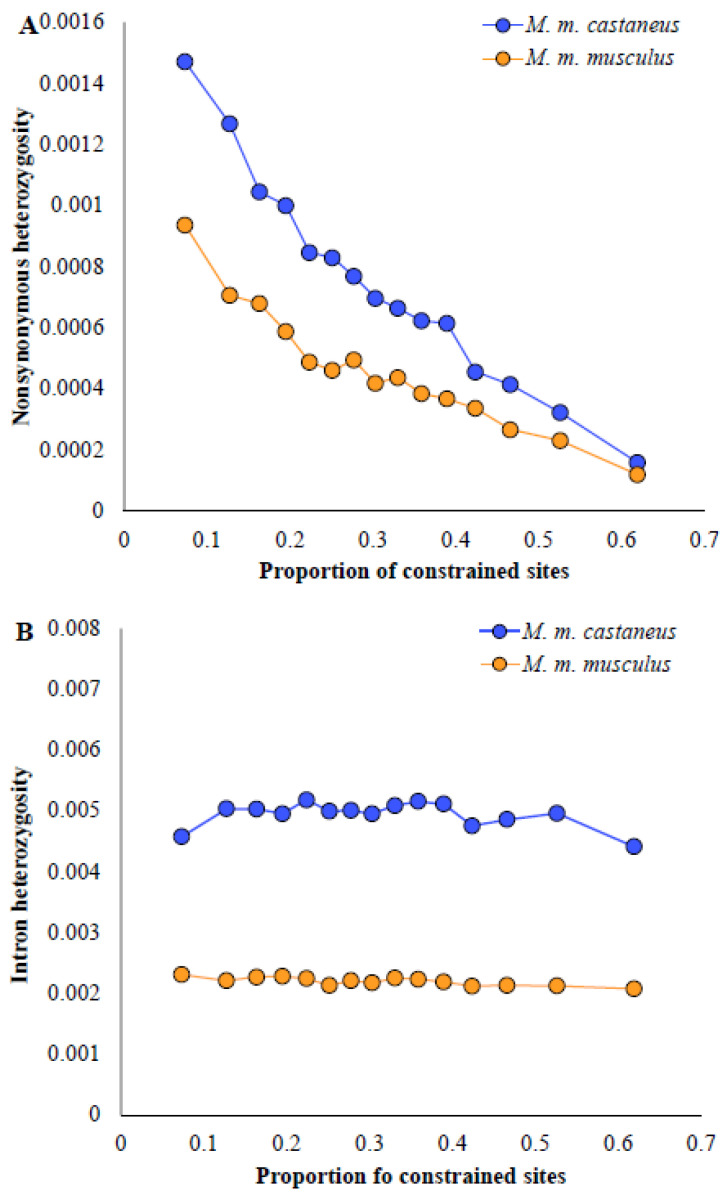
The mean proportion of constrained sites obtained for genes belonging to 15 categories was plotted against their heterozygosity at (**A**) nonsynonymous sites and (**B**) introns. A total of 14,870 genes were grouped into 15 categories based on the proportion of their sites with a PhyloP score > 2.0.

**Figure 4 biology-13-00810-f004:**
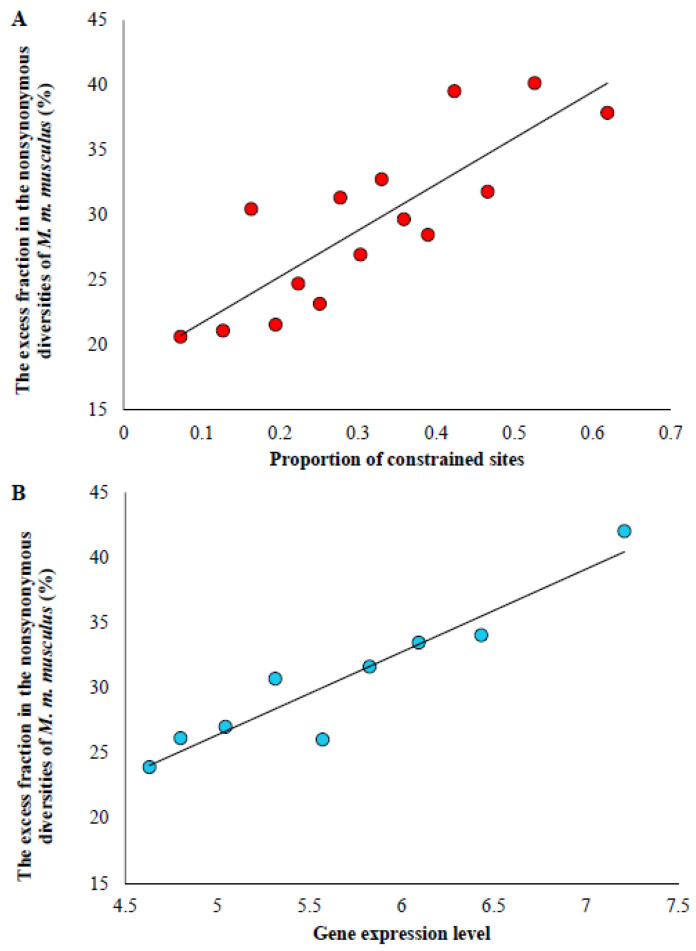
(**A**) Correlation between the mean proportion of constrained sites and magnitude of excess fraction in the nonsynonymous diversities of *M. m. musculus* for genes belonging to 15 categories based on the level of selective constraints on them. These two variables have a highly significant positive correlation (*r* = 0.83, *p* = 0.00019). (**B**) Relationship between the level of gene expression and magnitude of excess fraction in the nonsynonymous diversities of *M. m. musculus* for genes belonging to nine expression level categories. This correlation was highly significant (*r* = 0.90, *p* = 0.002). The 9089 genes were grouped into nine categories based on their mean expression level.

## Data Availability

This study did not generate any new data. The chimpanzee genome data used in this study were obtained from https://eichlerlab.gs.washington.edu/greatape/data/VCFs/SNPs/ (accessed on 29 September 2023), and the mouse data are available from https://doi.org/10.5061/dryad.66t1g1k1j (accessed on 29 September 2023). In-house Perl scripts used in this study will be available upon request.

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
