# Peer review of "Purifying Selection Influences the Comparison of Heterozygosities between Populations"

_biology, 2024, doi:10.3390/biology13100810_

Round 1
Reviewer 1 Report
Comments and Suggestions for Authors
The article presents an analysis of one of the fundamental problems of population genetics. On the one hand, this is a classic standard study and does not contain any novelty, on the other hand, such an article can be useful for geneticists, zoologists, ecologists, especially for beginners as an example of such work.
However, when studying the manuscript, I had a number of questions:
1) After reading the manuscript, I came to the conclusion that the main goal of the article was to present an improved method for comparing populations of different species according to the specified genetic traits. In this regard, I recommend changing the title of the article. The article is methodological.
2) The author indicates previous works as material for the study [6, 19]. However, in these articles he is not a co-author. And in this manuscript, he is the only author. It is not entirely clear whose material the author used. If these are data posted in GenBank, this should be indicated.
3) The author did not provide any links to the data in GenBank, nor an appendix containing the sequences he worked with.
4) The use of the pronoun "we" when there is only one author is a bit confusing.
5) The description of the statistical analysis raises questions. First, what software was used to perform the analysis? If the author wrote codes for the analysis, they should be provided in the attachment. Second, what method did the author use to estimate mean pairwise differences? How many permutations were used in the bootstrap? The description of the correlation analysis is very trivial, this can be skipped. It would be better to provide the formula for the linear regression model (correlation analysis is a case of linear regression, although with nuances, but this is how it can be formulated).
6) Due to the unclear description of the statistical analysis and the simplest bar graphs, all the conclusions provided are not strongly supported by facts.
7) The figures are not of very good quality. The author provided the same figures in the appendix, but it would be better to place them in the text in good quality.
8) If the main objective of the article was to present an improved method for comparing populations of different species by specified genetic traits, then the methodology, including statistical analysis, should be described more clearly. For a more visual presentation, it is better to provide a study scheme and add an appendix/links to the data in and add an appendix/links to the data in GenBank.
Author Response
Comments: After reading the manuscript, I came to the conclusion that the main goal of the article was to present an improved method for comparing populations of different species according to the specified genetic traits. In this regard, I recommend changing the title of the article. The article is methodological.
If the main objective of the article was to present an improved method for comparing populations of different species by specified genetic traits, then the methodology, including statistical analysis, should be described more clearly.
Response: I have now rewritten the paper to focus more on the findings rather than the methodology.
Comments: The author indicates previous works as material for the study [6, 19]. However, in these articles he is not a co-author. And in this manuscript, he is the only author. It is not entirely clear whose material the author used. If these are data posted in GenBank, this should be indicated.
The author did not provide any links to the data in GenBank, nor an appendix containing the sequences he worked with.
For a more visual presentation, it is better to provide a study scheme and add an appendix/links to the data in and add an appendix/links to the data in GenBank.
Response: I have now provided the exact link to the dataset in the methods and data availability sections.
Comment: The use of the pronoun "we" when there is only one author is a bit confusing.
Response: I have changed the whole manuscript in passive voice to avoid the confusion.
Comments: The description of the statistical analysis raises questions. First, what software was used to perform the analysis? If the author wrote codes for the analysis, they should be provided in the attachment. Second, what method did the author use to estimate mean pairwise differences? How many permutations were used in the bootstrap? The description of the correlation analysis is very trivial, this can be skipped. It would be better to provide the formula for the linear regression model (correlation analysis is a case of linear regression, although with nuances, but this is how it can be formulated).
Due to the unclear description of the statistical analysis and the simplest bar graphs, all the conclusions provided are not strongly supported by facts.
Response: I have provided the details on the number of bootstrap replicates; method used to calculate pairwise heterozygosity and clarified the type of correlation analysis performed in the methods section.
Comment: The figures are not of very good quality. The author provided the same figures in the appendix, but it would be better to place them in the text in good quality.
Response: We have provided good quality figures now and will provide with 300 dpi pictures for the publication.

Reviewer 2 Report
Comments and Suggestions for Authors
Good article. Just minor corrections (noted in the file).

Author Response
Comment: This appears to be the opposite since the population of M. m. castaneous is larger than that of M. m. musculus.
Response: Thanks for pointing this out. I have corrected this.
Comments: This sentence may lead to an interpretation that the value is 2.26 times greater than the value for M. m. musculus (i.e., 3.36 times the value for M. m. musculus). To avoid this misinterpretation, maybe its better to write "2.26 times as high as" ou "2.26 times the value for M. m. musculus". The same is true for the next two statements
Response: I have rewritten these statements to avoid any misinterpretation.
Comment: I didn't find this information previously in the article. Furthermore, it seems to me that this information would be better included in the Results and/or Discussion.
Response: I found it difficult to incorporate these statements in the results section without disturbing the flow.
Reviewer 3 Report
Comments and Suggestions for Authors
The authors provide a method for (or a study of) neutral and constrained polymorphism in two populations (or subspecies) of the same species. In particular, the focus is on the effects of varying effective population sizes on the ratio of ratios of neutral and constrained polymorphism (both estimated via pi) in a large and a small population. To quantify this, the authors propose a metric, which they call "overestimation of heterozygosity". In statistics the term "overestimation" however suggests a bias when estimating a quantity. But estimates of pi are not (necessarily) biased. Rather the metric quantifies the excess over the expectation in the ratio, which makes the term "overestimation" misleading. Hence, I suggest that the authors call their metric "excess ..." instead of "overexpression ...". Furthermore, similar approaches and metrics have already been put forward, eg, the McDonald-Kreitman test that could also be framed as a deviation of a ratio of ratios from expectations. Hence, what the authors consider the main point (according to the title) does not seem quite novel to me.
Furthermore, the authors remain quite ambiguous about whether they assume neutral equilibrium and barely mention population demography. Non-equilibrium due to shrinking population size is mentioned around the term "spurious" in the discussion, but the reference is to Kimura's founding work and not to any more modern work. As an example for this modern work, take recent efforts to estimate Neandertal effective population sizes (Ne). To this topic the authors write in their introduction: "The higher pi_N/pi_S ratio observed in Neanderthals than that in modern humans was due to the drastic decline in the population of Neanderthals before their extinction [4, 5]." Consider however: Rogers (2024, 10.24072/pcjournal.448) who argue quite convincingly that population demography and sampling produce the observed pattern of apparent shrinking of Neanderthal Ne. Overall, I consider the theoretical population genetic foundation of the article rather weak, because many of the more recent efforts are ignored and the approach is not really new.
Switching to statistics: The authors write: "The standard error was estimated using the bootstrap method by resampling one mb genomic blocks." I do not think that the authors here mean the standard error but rather the standard deviation. Furthermore, I do not understand this sentence: "The regression analyses were performed to examine the correlation between the proportion of constrained sites or gene expression level and overestimation." A regression is different from a correlation, the latter being a descriptive statistic of the relationship between two variables, the former assumes a linear model. I guess in this case, it can be assumed that the proportion of constrained sites or gene expression level are the explanatory variables and the "overestimation" metric the target variable. Hence, a regression analysis would be more appropriate than a correlation analysis in this case. In any case, the authors should make their statistical model clear.
While I consider the population genetic theory and statistics rather weak, I consider the results of the data analyses quite interesting: the authors compare different classes of sites, eg introns, exons, highly and lowly expressed genes in two mammal species, mouse and chimpanzee. While I would urge more caution wrt the interpretation of the results (see the Neandertal example above), the authors' interpretations seem generally reasonable and plausible. Hence instead of putting the emphasis on the metric (to which the title points), I urge the authors to put it on the results. This change of focus would mean that the authors would need to quite completely re-write the manuscript.
Comments on the Quality of English LanguageThe quality of the language was good enough for me to understand with a bit of guessing but should nevertheless be improved.
Author Response
Comments: The authors provide a method for (or a study of) neutral and constrained polymorphism in two populations (or subspecies) of the same species. In particular, the focus is on the effects of varying effective population sizes on the ratio of ratios of neutral and constrained polymorphism (both estimated via pi) in a large and a small population. To quantify this, the authors propose a metric, which they call "overestimation of heterozygosity". In statistics the term "overestimation" however suggests a bias when estimating a quantity. But estimates of pi are not (necessarily) biased. Rather the metric quantifies the excess over the expectation in the ratio, which makes the term "overestimation" misleading. Hence, I suggest that the authors call their metric "excess ..." instead of "overexpression ...". Furthermore, similar approaches and metrics have already been put forward, eg, the McDonald-Kreitman test that could also be framed as a deviation of a ratio of ratios from expectations. Hence, what the authors consider the main point (according to the title) does not seem quite novel to me.
Overall, I consider the theoretical population genetic foundation of the article rather weak, because many of the more recent efforts are ignored and the approach is not really new.
While I consider the population genetic theory and statistics rather weak, I consider the results of the data analyses quite interesting: the authors compare different classes of sites, eg introns, exons, highly and lowly expressed genes in two mammal species, mouse and chimpanzee. While I would urge more caution wrt the interpretation of the results (see the Neandertal example above), the authors' interpretations seem generally reasonable and plausible. Hence instead of putting the emphasis on the metric (to which the title points), I urge the authors to put it on the results. This change of focus would mean that the authors would need to quite completely re-write the manuscript.
Response: I agree with the reviewer and thank them for pointing this out. Based on the suggestion, I have rewritten the whole paper right from the title to focus on the findings rather than the methodology development. Similarly, as suggested by the reviewer we changed the term from “overestimation” to “excess” throughout the manuscript. Furthermore, I have now mentioned about the principle of MK test and the studies that followed and extended this method (see methods).
Comments: Furthermore, the authors remain quite ambiguous about whether they assume neutral equilibrium and barely mention population demography. Non-equilibrium due to shrinking population size is mentioned around the term "spurious" in the discussion, but the reference is to Kimura's founding work and not to any more modern work. As an example for this modern work, take recent efforts to estimate Neandertal effective population sizes (Ne). To this topic the authors write in their introduction: "The higher pi_N/pi_S ratio observed in Neanderthals than that in modern humans was due to the drastic decline in the population of Neanderthals before their extinction [4, 5]." Consider however: Rogers (2024, 10.24072/pcjournal.448) who argue quite convincingly that population demography and sampling produce the observed pattern of apparent shrinking of Neanderthal Ne.
Response: I have now discussed about the alternative possibility suggested by Rogers in the discussion section.
Comments: Switching to statistics: The authors write: "The standard error was estimated using the bootstrap method by resampling one mb genomic blocks." I do not think that the authors here mean the standard error but rather the standard deviation. Furthermore, I do not understand this sentence: "The regression analyses were performed to examine the correlation between the proportion of constrained sites or gene expression level and overestimation." A regression is different from a correlation, the latter being a descriptive statistic of the relationship between two variables, the former assumes a linear model. I guess in this case, it can be assumed that the proportion of constrained sites or gene expression level are the explanatory variables and the "overestimation" metric the target variable. Hence, a regression analysis would be more appropriate than a correlation analysis in this case. In any case, the authors should make their statistical model clear.
Response: I agree and modified this. Since the nature of relationship between the variables is unknown (eg. linear or non-linear) we used the non-parametric Spearman’s rank correlation. This has been clarified in the methods section.